# Baricitinib: A 2018 Novel FDA-Approved Small Molecule Inhibiting Janus Kinases

**DOI:** 10.3390/ph12010037

**Published:** 2019-03-12

**Authors:** Annie Mayence, Jean Jacques Vanden Eynde

**Affiliations:** 1Haute Ecole Provinciale de Hainaut-Condorcet, 7330 Saint-Ghislain, Belgium; annie.mayence@alumini.umons.ac.be; 2Formerly head of the Department of Organic Chemistry (FS), University of Mons-UMONS, 7000 Mons, Belgium

**Keywords:** approved drugs, baricitinib, FDA, Janus kinases, protein kinase inhibitor, rheumatoid arthritis

## Abstract

In 2018, Baricitinib was approved by the Food and Drig Administration (FDA) for the treatment of rheumatoid arthritis. Baricitinib exerts its action by targeting Janus kinases (JAK). In this study, we describe the necessary steps for preparing the drug using two alternative routes.

## 1. Introduction

In May 2001, Food and Drig Administration (FDA) approved the first kinase inhibitor [1], imatinib (**1**, Figure 1; Gleevec^®^ from Novartis, Basel, Switzerland), for the treatment of chronic myeloid leukemia. [2] Since that year, more than 45 kinase inhibitors have been marketed, most of them in oncology.

In 2018, 59 novel drugs have been approved by FDA. Among them, 40 can be considered as small molecules, 16 are derived from amino acids, and 3 from nucleic acids. Interestingly, almost 25% of the approved small molecules acted as kinase inhibitors and predominantly as protein kinase inhibitors. The purpose of this review is to provide a series of specific information on one of those inhibitors: baricitinib (LY3009104, formerly developed by Incyte Co as INCB028050 and subject of a license agreement with Eli Lilly and Co in 2009).

## 2. Baricitinib

### 2.1. Names and Structure

Baricitinib (**2**, Figure 2) is the active ingredient of Olumiant^®^, commercialized by Eli Lilly and Co. Its IUPAC name is: 2-[1-(ethanesulfonyl)-3-(4-{7*H*-pyrrolo[2,3-d]pyrimidin-4-yl}-1*H*-pyrazol-1-yl)azetidin-3-yl] acetonitrile, CAS 1187594-09-7.

### 2.2. Uses

After a rejection in April 2017, baricitinib (2 mg tablets) has been approved on May 31, 2018 for treatment of rheumatoid arthritis. [3] Noticeably, it had been approved, for the same purpose, in the European Union (EU) in February 2017. [4]

### 2.3. Targets

Janus kinases (JAK) are tyrosine kinases (TYK) that play a crucial role in cell signaling [5,6]. They can be divided into four families: JAK1, JAK2, JAK3, and TYK2, and constitute interesting therapeutic targets [5,7]. The first JAK inhibitor approved by FDA (November 2011) was ruxolitinib (**3**) [8]. Through its selective inhibition of JAK1 and JAK2 (Table 1), it is clinically used for intermediate or high-risk myelofibrosis.

Baricitinib (**2**) is also a selective and reversible inhibitor of JAK1 and JAK2 with less affinity for JAK3 and TYK2 (Table 1). Interestingly, tofacitinib (**4**, Figure 2), another FDA-approved kinase (November 2012) used to treat rheumatoid arthritis [9], is even more selective (Table 1). 

Selectivity of inhibitors within the Janus kinases has been tentatively correlated to specific interactions (hydrogen bonds) with amino acid residues in the hinge region of the ATP binding site [5]. Studies started during a screening of approximately 400,000 compounds from a Pfizer library in order to discover an inhibitor of JAK3. This enabled the identification of 9-(7*H*-Pyrrolo[2,3-d]pyrimidin-4-yl)-2,3,4,4a,9,9a-hexahydro-1*H*-carbazole (**5**) as a lead, which was reported by Flanagan et al. [10]. Improvements of the properties of 5, among which its metabolic stability [10] led to the identification and development of commercialized drugs **2**-**4**.

### 2.4. In Vitro Studies, Rodent Models, and Clinical Trials

Several proinflammatory cytokines are involved in the pathogenesis of rheumatoid arthritis. Mention can be made of interleukin (IL) 6, IL-15, IL-17, IL-23, interferon-α/β, interferon-γ, and granulocyte-macrophage colony as stimulating factors. [10] As elegantly depicted by Furumoto and Gadina [11], such activity is critically linked to JAK signaling pathways and the signal transducer and activator of transcription (STAT) signaling pathways. Therefore, targeting those pathways represented and still represents a challenging field of research [11,12].

In an in-depth preclinical study performed by Incyte Co, Fridman et al. [13], they reported that the action of baricitinib in peripheral blood mononuclear cells (PBMCs) could prevent the production of pathogenic and proinflammatory cytokines. That production was not altered by structural analogs that did not inhibit JAK1 and JAK 2. Baricitinib has been administered orally (1, 3, and 10 mg/kg/day) or by constant infusion in several rodent models (rats and mice). Clinical, histologic, radiographic, and hematologic data demonstrated the efficacy and safety of the drug, thus justifying clinical trials.

Following the NIH [14], baricitinib has been the subject of 30 completed clinical trials, namely 17 phase 1 studies, 7 phase 2 studies, and 6 phase 3 studies; 20 other trials are ongoing or scheduled. Historically, the first trial was a phase 2 study launched on May 15, 2009. It was entitled “INCB028050 Compared to Background Therapy in Patients with Active Rheumatoid Arthritis (RA) with Inadequate Response to Disease Modifying Anti-Rheumatic Drugs” (NCT00902486) and was summarized as follows: “This was a randomized, double blind, placebo controlled, dose ranging, parallel group study. Participants who had active rheumatoid arthritis (RA) who had inadequate response to any disease modifying anti-rheumatic drug (DMARD) therapy including biologics were enrolled. Screening evaluations were performed within approximately 28 days of randomization. The duration of the study was [six] months with the primary endpoint assessed at [three] months. Eligible participants were randomly assigned to one of three doses (4, 7, or 10 mg QD) of INCB028050 (Baricitinib) or placebo.” The first phase 3 studies were initiated at the end of 2012 under the titles “A Study in Moderate to Severe Rheumatoid Arthritis (RA-BEAM)” (NCT01710358; first posted October 19, 2012), “A Study in Participants with Moderate to Severe Rheumatoid Arthritis (RA-BEGIN)” (NCT01711359; first posted October 22, 2012), “A Moderate to Severe Rheumatoid Arthritis Study (RA-BEACON)” (NCT01721044; first posted November 2, 2012), and “A Study in Moderate to Severe Rheumatoid Arthritis Participants (RA-BUILD)” (NCT01721057; first posted November 2, 2012). All results confirmed the high efficiency of baricitinib and underlined a limited incidence of side effects such as a decrease in hemoglobin and an increase in LDL, HDL, creatinine, and creatine phosphokinase. Further details on clinical trials can be found in references [14,15,16,17,18,19].

### 2.5. Syntheses

There are essentially two routes for the preparation of baricitinib **2**. As depicted in Scheme 1, they can be distinguished by introducing central pyrazole ring in the molecule. In the original procedure [20,21], the pyrazole ring was linked to the pyrrolo[2,3-d]pyrimidine system (to afford **6**) and then coupled to the azetidine moiety **7** to give the intermediate **8**. In an alternative route [22,23], the bound between the pyrazole and the azetidine was formed (to yield **10**) before reaction with the fused system **9**.

Thus (Scheme 2), 4-chloro-7*H*- pyrrolo[2,3-d]pyrimidine was protected on position 7 by reaction with 2-(trimethylsilyl)ethoxymethyl chloride. The protected fused system was then coupled with 4-pyrazoleboronic acid pinacol ester **12** by a Suzuli-Miyaura reaction, giving **6**. Parallelly, **7** was obtained from 1-Boc-3-azetidinone **13** and diethyl cyanomethylphosphonate. Reaction between **6** and **7** in the presence of DBU afforded the ester **8**. Subsequent hydrolysis, decarboxylation, sulfonation, and finally deprotection of the pyrrolopyrimidine moiety yielded the targeted derivative **2**. In a variant [21], also used to prepare deuterated samples of **2** [24], the azetidine derivative **7** has been deprotected and sulfonated before coupling with **6**.

In a more recent patent [22], the sulfonated azetidine **14** (Scheme 3) was prepared from azetidine-3-ol by a sequence including a sulfonation, an oxidation, and introduction of the cyanomethylene moiety. Interestingly, there is no need to protect any position in that sequence. Additionally, let us emphasize that the oxidation step could be performed both in batch or under flow conditions. [22,25]. Then, **14** was reacted with 4-pyrazoleboronic acid pinacol ester **12** to yield **10**. The bound between the azetidinylpyrazole group and the pyrrolo[2,3-d]pyrimidine system was then created through a Suzuki-Miyaura reaction involving 7-Boc-4-chloro-7H-pyrrolo[2,3-d]pyrimidine **9** or even the unprotected 4-chloro-7*H*-pyrrolo[2,3-d]pyrimidine.

## 3. Perspectives

As expected, success of the 7*H*-pyrrolo[2,3-b]pyrimidine scaffold in the development of drugs for treatment of various diseases and essentially rheumatoid arthritis has initiated many researches on structurally related analogs. Among them, the 1H-pyrrolo[2,3-b]pyridine skeleton emerged as a moiety of particular interest, as indicated by the number of recent publications and patents in which it has been described [26,27,28,29,30,31].

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
