# Peer review of "Baricitinib: A 2018 Novel FDA-Approved Small Molecule Inhibiting Janus Kinases"

_pharmaceuticals, 2019, doi:10.3390/ph12010037_

Round 1

Reviewer 1 Report

The manuscript by Annie Mayence and co-workers deals with baricitinib, a JAK kinase inhibitors recently approved by FDA for the treatment of rheumathoid arthritis. The functional activity of the compound is described, together with the chemical route pursued to obtain it. The work is clear and well written, although a careful revision of the text is highly recommended to amend typos and improve the syntax.

However, although claimed as a ‘brief article’, the manuscript is actually too brief and general. Further information on the functional efficacy of the compounds should be added, like in vitro studies and, above all, clinical studies that allowed FDA approval. Moreover, the synthetic procedure for the obtainment of baricitinib should be explained and commented in details, and supplemented by a fully reaction scheme.

This points are mandatory for publication.

Author Response

We thank the reviewer for having given us the opportunity to strengthen our manuscript.

A paragraph has been added dealing with in vitro studies, rodent models, and clinical trials. Nine supportive references have been added.

Two alternative routes to baricitinib has been developed as suggested by the reviewer. Two schemes have been added,

All changes have been highlighted in yellow. 

Reviewer 2 Report

In this Brief report, the authors explain the key steps of two synthesis of Baricitinib.

This is an interesting molecule which has been the subject of several patents.

However, the discussion in this manuscript is too brief. In fact, the strategies are quite similar. The two key reactions (addition of the pyrazolyl unit to the azetidine fragment and coupling reaction) are used in both synthesis albeit in different order.

There are other recent reports on the improvement of some steps or the preparation of intermediates.

Thus, for this report to be useful, some more information should be given, general strategies used, differences between the most relevant processes, improvements introduced…

With some more information, this report could be of interest to the medicinal chemistry community.

Author Response

As suggested by the reviewer, the two alternative routes to baricitinib have been developed and the corresponding schemes have been added,

All changes have been highlighted in yellow. 

Reviewer 3 Report

It is a short overview but it is much too short because of shortage of information. Authors wrote in the abstract: "Key step for the preparation of the drug by two alternate routes is desribed." But indeed, they are NOT described. In my opinion, steps of synthesis, which are marked in enigmatic way and they should be described detaily. The preparation of compounds 6-9 should be explained too. I would suggest to develop a bit this fragment of the manuscript and it will be suitable for publication then.

Chers Auteurs, votre article est interessant et bien ecrit, mais, quand-meme, apres ayant lu cet article, je voudrais bien savoire un peu plus au sujet des methodes de synthese du baricitinib

Author Response

(The authors gave the same response as above.)

Round 2

Reviewer 1 Report

The authors deeply revised the manuscript, improving its content by adding functional studies on the compounds and synthetic procedures for its obtainment. They fully acknowledge the reviwer's suggestions, thus making the manuscript ready for publication.

Reviewer 2 Report

The manuscript is now more interesting as morerelevant information is presented.

The synthetic part has also been improved according to the suggestions.